# Aqueous Dried Extract of *Skytanthus acutus* Meyen as Corrosion Inhibitor of Carbon Steel in Neutral Chloride Solutions

**Luis Cáceres** [1,*] , **Yohana Frez** [1] , **Felipe Galleguillos** [1] , **Alvaro Soliz** [2] , **Benito Gómez-Silva** [3] and **Jorge Borquez** [4]

1. Departamento de Ingeniería Química y Procesos de Minerales, Universidad de Antofagasta, Antofagasta 1240000, Chile; yohana.frez.segovia@ua.cl (Y.F.); felipe.galleguillos.madrid@ua.cl (F.G.)
2. Departamento de Ingeniería en Metalurgia, Universidad de Atacama, Copiapó 1530000, Chile; alvaro.soliz@uda.cl
3. Laboratorio de Bioquímica, Departamento Biomédico, Facultad Ciencias de la Salud and Centre for Biotechnology and Bioengineering (CeBiB), Universidad de Antofagasta, Antofagasta 12701300, Chile; benito.gomez@uantof.cl
4. Departamento de Química, Universidad de Antofagasta, Antofagasta 1240000, Chile; jorge.borquez@uantof.cl
* Correspondence: luis.caceres@uantof.cl

**Abstract:** The implementation of corrosion engineering control methods and techniques is crucial to extend the life of urban and industrial infrastructure assets and industrial equipment affected by natural corrosion. Then, the search of stable and environmentally friendly corrosion inhibitors is an important pending task. Here, we provide experimental evidence on the corrosion inhibitory activity of aqueous extracts of *Skytanthus acutus* Meyen leaf, a native plant from the Atacama Desert in northern Chile. *Skytanthus* extracts as a powder should be prepared at 55 °C to avoid thermal decomposition and loss of corrosion inhibitory activity. Corrosion of carbon steel AISI1020 immersed in 0.5 M NaCl was evaluated in the presence of different doses of *Skytanthus* extract by complementary and simultaneous linear polarization, electrochemical impedance spectroscopy, and weight-loss technique under high hydrodynamic conditions. Mixed Potential Theory was applied to confirm the electrochemical activity of the extract inhibitory capabilities. The *Skytanthus* extracts reached a 90% corrosion inhibitory efficiency when tested at 100 to 1200 ppm in a time span of 48 h, through an electrochemical interaction between the extract inhibitor component and the carbon steel surface. The corrosion inhibition activity observed in *Skytanthus* dry extracts involves a protective film formation by a mechanism that includes an iron dissolution at the expense of either oxygen reduction and/or hydrogen evolution, followed by a ferrous-ferric iron cycling, the formation of an iron complex and adsorption to the metal surface, and, finally, desorption or degradation of the protecting film. The water-soluble plant extract was subjected to HPLC-MS analyses that rendered 14 major signals, with quinic acid, protocatechuic acid, chlorogenic acid isomers, vanillic acid hexoside, and patuletin 3-methoxy-7-glucoside as the most abundant components. Then, we propose that a phenolic derivative is responsible for the corrosion inhibitory activity found in *Skytanthus* extracts.

**Keywords:** *Skytanthus acutus* Meyen; carbon steel; EIS; weight loss; corrosion inhibition; inhibition mechanism

## 1. Introduction

Although carbon steel has no capacity to develop a protective surface oxide layer against the aggressiveness of the surrounding environment, it is the most widely used steel material in engineering applications. This is justified by its significant lower cost in comparison to more noble higher-grade alloys, mechanical properties, and its amenability to be safely operated in very corrosive service conditions under a properly designed and implemented control system. In such cases, corrosion inhibitors are added in very small

amounts that adhere to the metal surface to form a protective barrier against corrosive agents contacting the metal. The efficiency of an inhibitor to provide corrosion protection depends to a large extent upon the interactions between the inhibitor and the metal surface [1]. The use of corrosion inhibitors for carbon steel in chloride-containing solutions is steadily increasing mainly because the seawater transport and processing is becoming the main alternative to cope with the expanding trend of freshwater scarcity. This is particularly true in mineral extracting and processing plants around the world which are facing decreasing lower-grade ores and increasing water consumption rates [2,3]. In addition, mineral processing plants in northern Chile also require pumping of seawater from seashores to sites as high as 3000 m above sea level. Thus, the use of corrosion inhibitors must comply with relevant requirements, such as high inhibition efficiency under severe hydrodynamics conditions. Specific testing standards for such conditions exist; for example, a weight-loss test which considers a steel specimen (ASTM 2688-05) exposed to linear velocities up to 2 m/s inside horizontal pipe in a water recirculating system. Given the logistic tasks associated with such standard practice, this method is not normally used at the early corrosion inhibitor research testing; instead, linear sweep voltammetry, cyclic voltammetry, and electrochemical impedance spectroscopy are electrochemical methods mainly preferred which sometimes are complemented with weight-loss testing which consists of immersion of test coupons on quiescent solutions for long predetermined times [4–6].

A number of inorganic inhibitors for carbon steel in chloride media have been found and widely investigated. Chromates are the most effective compounds to have been investigated as corrosion inhibitors for different metals and alloys, including mild steel, carbon steel, aluminum, zinc, copper, and their intermetallic alloys in chloride media. High toxicity and adverse impact on the environment do not permit the use of chromate as corrosion inhibitor [7]. Zinc gluconate, zinc acetate, and zinc acetylacetonate are known to be efficient inhibitors among them, and zinc gluconate has shown the best performance [8]. Recently hydrophobic organosilicon dispersions have been investigated as inhibitor for carbon steel in low-concentration chloride media; active molecules chemisorb on the surface of metal producing a maximum 85% inhibition efficiency [9].

Low toxicity, eco-friendliness, and good efficiency are important aspects to consider for novel corrosion inhibitors to replace conventional synthetic inorganic or organic inhibitors which, although they have shown effective corrosion inhibition, their cost, toxicity, and non-biodegradability are significant drawbacks.

Corrosion inhibitors from plant extracts have recently been reported [10,11]; most of them focused on metals in acidic corrosive media. The only references found for corrosion inhibition in chloride neutral solutions are for neem leaves [12], rice straw [6], and seaweed extract [13]. Among these studies the maximum corrosion inhibition efficiency under quiescent conditions was 90% for a seaweed extract dose of 1.2% (*v/v*); however relevant aspects, such as product stability, results reproducibility, and hydrodynamic corrosion effects, were not mentioned in these reports.

Interestingly, reports on the corrosion protection of steel-reinforced concrete immersed in NaCl solution are available [14]; for example, the effect of extracts from *Phyllanthus muellerianus* leaves in concrete immersed in 3.5% NaCl, simulating a saline/marine environment [15]. Despite the good corrosion inhibition efficiency values reported it is unlikely that these results could be replicated for corrosion inhibition in steels immersed in corrosive media, because its validity is restricted to static corrosive media delimited by a porous matrix.

*Skytanthus acutus*, locally referred to as "goat's horn" for its seed pod shape, is an endemic shrub [16] from the hyperarid desert of northern Chile adapted to survive from occasional morning fog that occurs because of wide thermal daily fluctuations [17]. Skytanthine alkaloids are a group of natural oils of rare monoterpenoids isolated by a two-step steam distillation process from *Skytanthus acutus*, consisting of sequential steps of extraction in boiling methanol, vacuum concentration, filtration, and extraction with benzene [18,19].

From detailed skytanthine biosynthesis studies, the variation of enzymatic activity with plant age seems to be a natural phenomenon affecting *Skytanthus acutus.* Also, careful isolation work demonstrated the formation of several skytanthine isomers whose composition is non-uniformly distributed in different parts of the plant.

In this report we inform the successful inhibitory effect of dried aqueous extracts of *Skytanthus acutus* leaves on the corrosion of carbon steel as a rotating specimen in 0.5 M NaCl solution. We also provide an inhibition mechanism in terms of partial electrochemical reactions based on a combination of temporal measurement of electrochemical and weight-loss techniques for an electrode rotating at 1200 rpm. For all we know, no such systematic analysis about inhibitor efficiency studies in a rotating electrode has been performed before in which both electrochemical techniques and weight-loss measurement is combined in the same experimental run.

## 2. Materials and Methods

### 2.1. Dried Extracts

Fresh leaves of *Skytanthus acutus* (250 g) were collected from a coastal site with relatively abundant fog events, nearly 10 km south of Antofagasta City. One liter of water-soluble extracts was obtained after 10 min blender grinding in distilled water, at room temperature, followed by vacuum filtration. The extract was evaporated to a 90% volume reduction in an open pan under magnetic stirring with forced air flow from an electric fan located at 10 cm over the liquid surface. A dried residue (DR) was obtained after complete evaporation in an oven (Binder GmbH, Tuttlingen, Germany) for 5 d. To avoid loss of the anticorrosion activity of the plant extracts, temperature was maintained at 55 °C in all the evaporation steps conducted.

### 2.2. Aqueous Extract General Characterization

The identification of metabolites in the aqueous extract was performed using an ultra-high-performance liquid chromatography–tandem mass spectrometry and a photodiode array detector referred as UHPLC-PDA-HESI-orbitrap-MS/MS (Thermo Fisher Scientific, Bremen, Germany). The detection wavelengths were 330, 280, 254, and 354, and photodiode array detectors were set from 200–800 nm. The Q-exactive 2.3 SP 2, Xcalibur 2.4, and Trace Finder 3.3 (Thermo Fisher Scientific, Bremen, Germany) were used for UHPLC mass spectrometer control and data processing, respectively. The equipment description and parameter adjustments are described elsewhere [20,21]. Solvent delivery was performed at 1 mL/min using ultra-pure water supplemented with 1% formic acid (A) and acetonitrile with 1% formic acid (B) and a program starting with 5% B in 10 min. This condition was kept for additional 12 min to achieve column stabilization before sample injection. For the analysis, 20 μL of aqueous extract was injected in the instrument. All these reagents were provided Merck S.A., Santiago, Chile.

Sixteen compounds were detected in UHPLC-PDA-OT-MS analysis, combining full mass spectra and MS experiments, 14 of which were tentatively identified, including organic acid such as quinic acid, chlorogenic acid, ascorbic acid, citric acid, benzoic acid glycosidases, flavonoids, and flavonoid glycosidases.

### 2.3. Electrochemical Tests

The electrochemical measurements were performed using a conventional three-electrode setup mounted in a home-made rotating electrode accessory connected to an SP1 Zive Potentiostat (ZiveLab, WonA Tech, Seoul, Korea). The working electrode (WE) was a carbon steel specimen, the reference electrode an Ag/AgCl (KCl Sat.) half-cell, and the counter electrode either a platinum coil or a graphite rod. The steel used in all experiments was commercial carbon steel type AISI 1020, with a chemical composition (wt.%) of 0.2 C, 0.6 Mn, 98.5 Fe and traces of S, Si, Cu, Ni, Cr, Sn, P, and Mo. NaCl was supplied by Merck S.A. (Merck S.A., Santiago, Chile). All potentials referred to the standard hydrogen electrode (SHE).

Two different measurement cells were used:

- A 30 mL cell (SC) for linear sweep voltammetry (LSV) measurements under different DR dosing. The dosing was performed from a DR solution of 20 g/L of DR in 0.5 M NaCl. The LSV scan range was from $-1000$ to 0 mV/SHE at a scan rate of 2 mV/s. In this cell the WE were manufactured from a carbon steel rod with a diameter of 4.2 mm and a length of 10 mm. This was concentrically inserted using resin adhesive in a PTFE cylinder of 8 mm of diameter that serves a fixing device to the shaft of the rotating disk electrode cell stand (Figure 1a). Thus, the exposed area was a disk of $2.7 \times 10^{-5}$ m$^2$. The counter electrode was a platinum coil.

- A 200 mL cell (LC) adapted for a sequential measurement of LSV and electrochemical impedance spectroscopy (EIS). The WE were a carbon steel cylinder of 28 mm length and 13 mm diameter with a 9 mm threaded extension to be coupled to a motor shaft adaptor using an O-ring sealing, to keep the threaded section in dried condition while the exposed surface specimen is immersed in the electrolyte (Figure 1b). Before each test the WE were cleaned in an ultrasonic bath with distilled water then washed with isopropyl alcohol and acetone, dried at air, and finally weighed in an analytical scale. The exposed surface of this cylindrical WE were $1.27 \times 10^{-3}$ m$^2$. Using this WE design, LSV and EIS measurements were made at different immersion times along 48 h, combined with a global corrosion rate determination by the weight-loss method at the end of the run. For that purpose, the corroded WE were cleaned according to ASTM G1-90 to be weighed in an analytical scale.

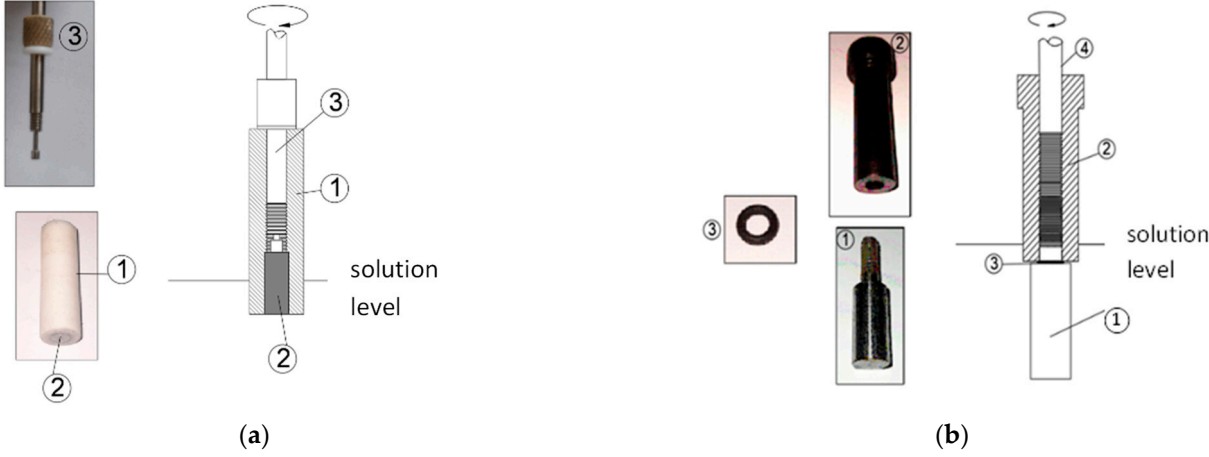

(**a**)　　　　　　　　　　　　　　　　　　　　　　　　　　　　(**b**)

**Figure 1.** (**a**) WE for SC cell: 1. PTFE cylinder, 2. AISI 1020 specimen embedded in the PFTE cylinder, and 3. Rotator shaft with threaded connection and retractable rod for electric connectivity. The exposed surface is the frontal disk; (**b**) WE for LC cell: 1. AISI 1020 cylindrical specimen with threaded connector, 2. PTFE holder with threaded connection to specimen and rotating shaft, 3. sealing O-ring, and 4. Rotator shaft with threaded connection and retractable rod for electric connectivity. The exposed surface is the disk surface and the cylinder delimited with the O-ring.

Three sets of experiments were conducted: the first two were carried out under N$_2$ or air bubbling, the WE rotated at 1200 rpm, and LSV, EIS, and weight loss were measured; the third group was done under different rotation rates and corrosion rates were measured using the weight-loss method. The oxygen content of the solutions was limited to that from air bubbling aeration at 20 °C. Nitrogen bubbling was applied to conduct experiments in the absence of oxygen.

The complete sequence of experimental steps is represented in Figure 2.

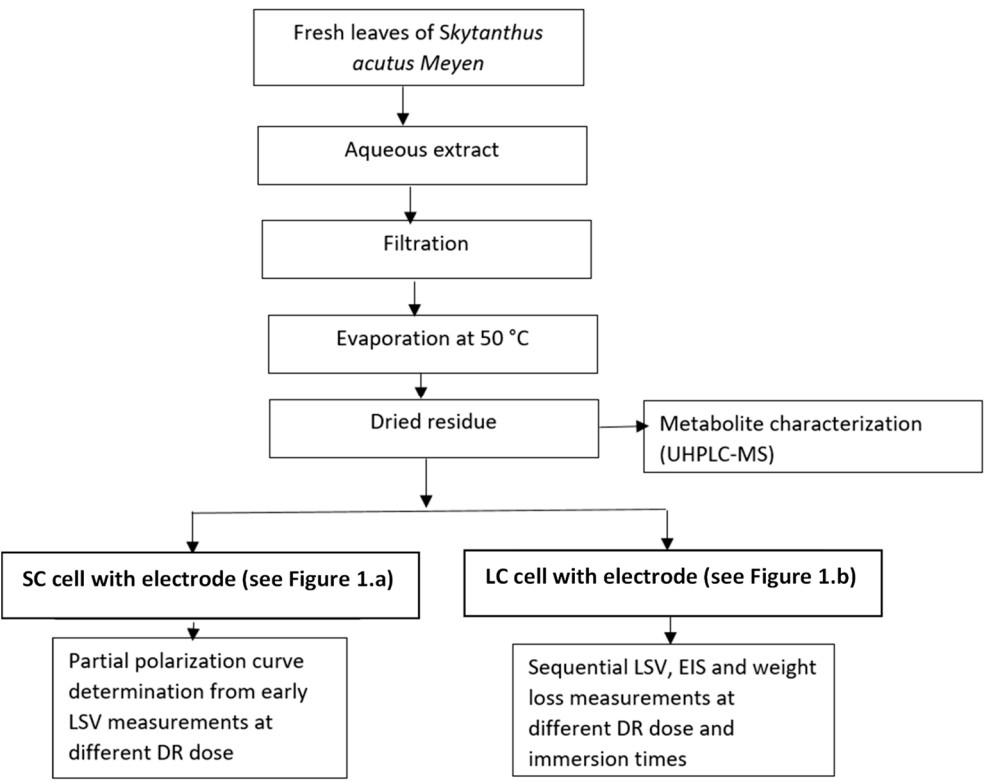

**Figure 2.** Sequence of experimental steps.

*2.4. Theory and Calculations*

2.4.1. Electrochemical Parameters from LSV Data

Carbon Steel in Pure NaCl Solutions

Earlier investigations have demonstrated that the potential mixed theory can be successfully applied to potential-current polarization data for iron in NaCl solutions assuming that the measured current density can be linearly decomposed in terms of partial electrochemical reactions of hydrogen evolution (HE), dissolved oxygen reduction (ORR), and iron oxidation (IO) [22,23]. The total current density, $I$, is expressed in terms of $I_{O_2}$, $I_{H_2}$ and $I_{Fe}$ the partial reduction current densities for ORR, HE, and IO, respectively (see Equation (A1) in the Appendix A).

Although values for $I_{O_2}$, $I_{H_2}$, and $I_{Fe}$ cannot be experimentally measured, they can be inferred, assuming kinetics expressions for each one of them. While the partial reactions for HE and IO follows a charge transfer kinetic mechanism, the ORR requires a kinetic expression for a mixed mechanism of charge transfer and diffusion control since it is significantly affected by oxygen mass transfer restrictions in liquid phase. The expressions for each partial reaction is an explicit equation [24] in terms of $I_{0,O_2}$, $I_{0,H_2}$, and $I_{0,Fe}$ the exchange current densities for ORR, HE, and IO, respectively, $I_l$ the limiting current density for ORR, $\eta_{O_2} = E - E_{eqO_2}$, $\eta_{H_2} = E - E_{eqH_2}$, and $\eta_{Fe} = E - E_{eqFe}$ the ORR, HE, and IO overpotentials, respectively, $E_{eqO_2}$, $E_{eqH_2}$, and $E_{eqFe}$ the equilibrium potentials for ORR, HE, and IO, respectively, the $t_{cO_2}$, $t_{cH_2}$, the cathodic Tafel slopes for ORR and HE, respectively, and $t_{aFe}$ the anodic Tafel slope for IO (see Equations (A2)–(A4) in the Appendix A). For the numerical manipulation process toward parameter determination using experimental $E$, $I$ data, these equations can be expressed in a simplifying form (see Equations (A5)–(A7) in the Appendix A) so the equilibrium potential values are unnecessary, and the number of parameters are three reaction coefficients $a_{O_2}$, $a_{H_2}$, and $a_{Fe}$, two cathodic Tafel slopes $t_{cO_2}$, $t_{cH_2}$, one anodic Tafel slope $t_{aFe}$ and one limiting current density $I_l$. The exchange current densities can be expressed in terms of these parameters (see Equations (A8)–(A10) in the Appendix A).

Carbon Steel in DR Containing NaCl Solutions

In this case we can formulate this expression:

$$I^* = I^*_{O_2} + I^*_{H_2} + I_{Ox} + I^*_{Fe} \tag{1}$$

where $I^*$ is the total current density in the presence of inhibitor, $I^*_{O_2}$, $I^*_{H_2}$, and $I_{Ox}$ are the partial reduction current densities for ORR, HE, and a global redox reaction for plant extract component(s), respectively, and $I^*_{Fe}$ is the partial iron oxidation current density in the presence of DR.

The algebraic expressions for $I^*_{O_2}$, $I^*_{H_2}$, and $I^*_{Fe}$ are supposed to be the same as those for the case without inhibitor, but with different parameter values. An expression for $I_{Ox}$ could be inferred from experimental measurements.

### 2.4.2. Weight-Loss Experiments

Weight-loss measurements were carried out in an LC cell. The rotation of the driving motor was preset by using a DC motor controller. The inhibition efficiency (*IE*) is defined as follows:

$$IE = \frac{\left(I^o_{corr} - I^i_{corr}\right)}{I^o_{corr}} \times 100 \tag{2}$$

where $I^o_{corr}$ and $I^i_{corr}$ are the corrosion rates density values the absence and the presence of inhibitor, respectively.

## 3. Results and Discussion

### 3.1. Plant Extract Preparation

Preliminary observations showed that evaporation steps conducted at temperatures higher than 80 °C rendered dry extracts darker than that obtained at temperatures below 60 °C. Also, the DR prepared at 80 °C showed an anticorrosion activity significantly lower than DR obtained at 60 °C, measured as polarization E-I data for carbon steel in solution. As evidence of thermal decomposition and loss of inhibitory activity a working temperature of 55 °C was adopted for extract preparations.

Bioactive compounds, such as flavonoid, vitamins, protein, and antioxidant, are known to suffer from a thermal degradation process when exposed to high temperature over a long period, which is often the case during the extraction and powder, making process [25]. The kinetics of thermal degradation of polyphenols has been studied in various fruits and medicinal plants extracts in the framework of food engineering. It has been reported that thermal decomposition of total phenolic content (TPC) in sweet cherry caused a sequential reduction in TPC in the whole processing temperature range studied, with a variation of 46% at 70–90 °C and between 47 to 63% up to 120 °C during a 60 min heating compared with untreated extracts [26]. Green tea catchequin and theaflavins, which were added to various drinks, were degraded by at least 50% within the first month of storage at room temperature [27]. The activation energy derived from the kinetic constant for thermal degradation and the half-time degradation have been identified as key parameters for the kinetics of thermal degradation for phytochemicals; an extensive list of these parameters has been measured, for example, a decrease in half-time values from 781.8 min at 70 °C to 182.4 min at 90 °C for thermal degradation of cyanidin-3-glucoside extracted from blood orange [28]. Freeze-drying or lyophilization is one of the most used processes for the protection of thermosensitive and unstable molecules [29].

In view of these bibliographical evidence it is strongly advised to identify conditions of thermal stability of plants extracts in order to be used as corrosion inhibitors. In this respect, no references were found concerning this relevant issue in corrosion nihibition studies.

### 3.2. Early LSV Curves

Figure 3 shows measured polarization curves for carbon steel in 0.5 M NaCl at early immersion time without inhibitor under aerated and unaerated conditions, including syn-

thetized partial polarization curves for the aerated case. Two main observations are, that (a) the fitted synthetized partial polarization curves for aerated and unaerated conditions fit well to the experimental curve, and partial $I_{H_2}$ curves, and (b) the partial $I_{H_2}$ curves superimpose under aerated and unaerated conditions. This last observation was corroborated by the close values of $a_{H_2}$ and $t_{cH_2}$ parameters from the aerated and unaerated solutions (Table 1).

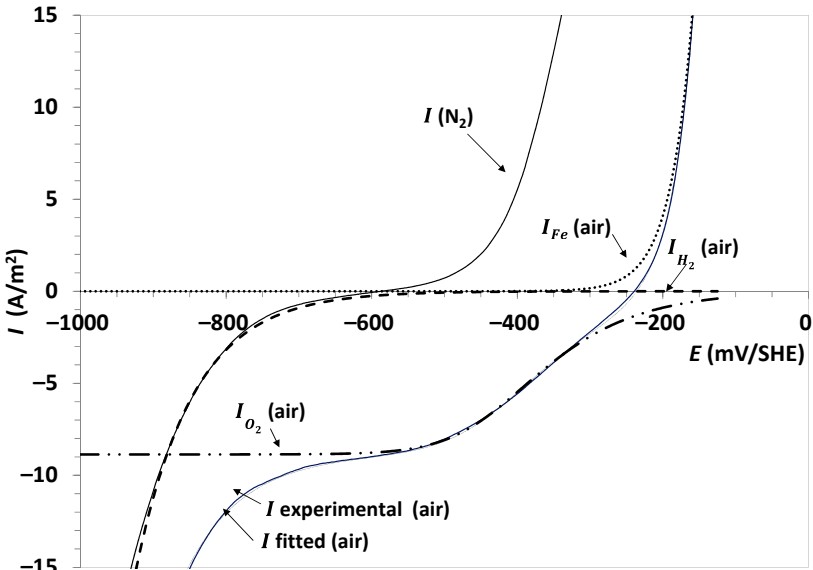

**Figure 3.** Polarization data for carbon steel in 0.5 M NaCl. Continuous lines: experimental data for aerated and unaerated conditions, with synthetized partial reactions as indicated.

**Table 1.** Electrochemical parameters for carbon steel in aerated and unaerated 0.5 M NaCl without inhibitor.

| Solution Condition | Kinetic and Corrosion Parameters | | | | | | | | |
|---|---|---|---|---|---|---|---|---|---|
| | $a_{O_2}$, A/m² | $a_{H_2}$, A/m² | $a_{Fe}$, A/m² | $I_l$, A/m² | $t_{cO_2}$, mV/dec | $t_{cH_2}$, mV/dec | $t_{aFe}$, mV/dec | $E_{corr}$, mV/SHE | $I_{corr}$, A/m² |
| Aerated | 0.1004 | $-1.33 \times 10^{-4}$ | 2524 | −8.9 | −206.2 | −184.7 | 71.8 | −237 | 1.24 |
| Unaerated | - | $-1.18 \times 10^{-4}$ | 12,308 | - | - | −181.1 | 119.5 | −585 | 0.16 |

Based on this fact it can be hypothesized that $I_{H_2}$ will not be altered in the presence of low concentration of DR, that is $I^*_{H_2} = I_{H_2}$. This is visually corroborated from a series of polarization curves at different DR dosing shown in Figure 4a. Under an increasing dosing trend the polarization curve evolves from 0 ppm as a gradual downward displacement while maintaining its cathodic exponential shape so that $I_{Ox}$ can be synthesized as $I^* - I_{H_2} - I^*_{Fe}$ (considering that the partial $I_{O_2}$ is null). The gradual downward displacement clearly indicates the presence of a partial reduction process that, in the absence of oxygen, it can be ascribed either to phenolics, terpenoids, polysaccharides, flavones, or derived compounds present in the plant extracts [30]. *Skytanthus acutus* contains a mixture of alkaloids of the rare monoterpenoid in various isomeric forms that have been partially characterized but whose biosynthetic pathways are not well understood [18,31]. The electrochemical activity in different substrates has been reported for only few of these compounds. In particular quinones undergo one-step two-electron reduction in aqueous buffer at acidic, neutral, and alkaline pH electrolytes [32] at potentials that could extend to negative values with an apparent hydrogen evolution reaction. Also, the antioxidant activity quantification of plant extracts by potentiometry or voltammetry based on the detection of a peak current in a nonreactive electrode is a well-known electrochemical method [33].

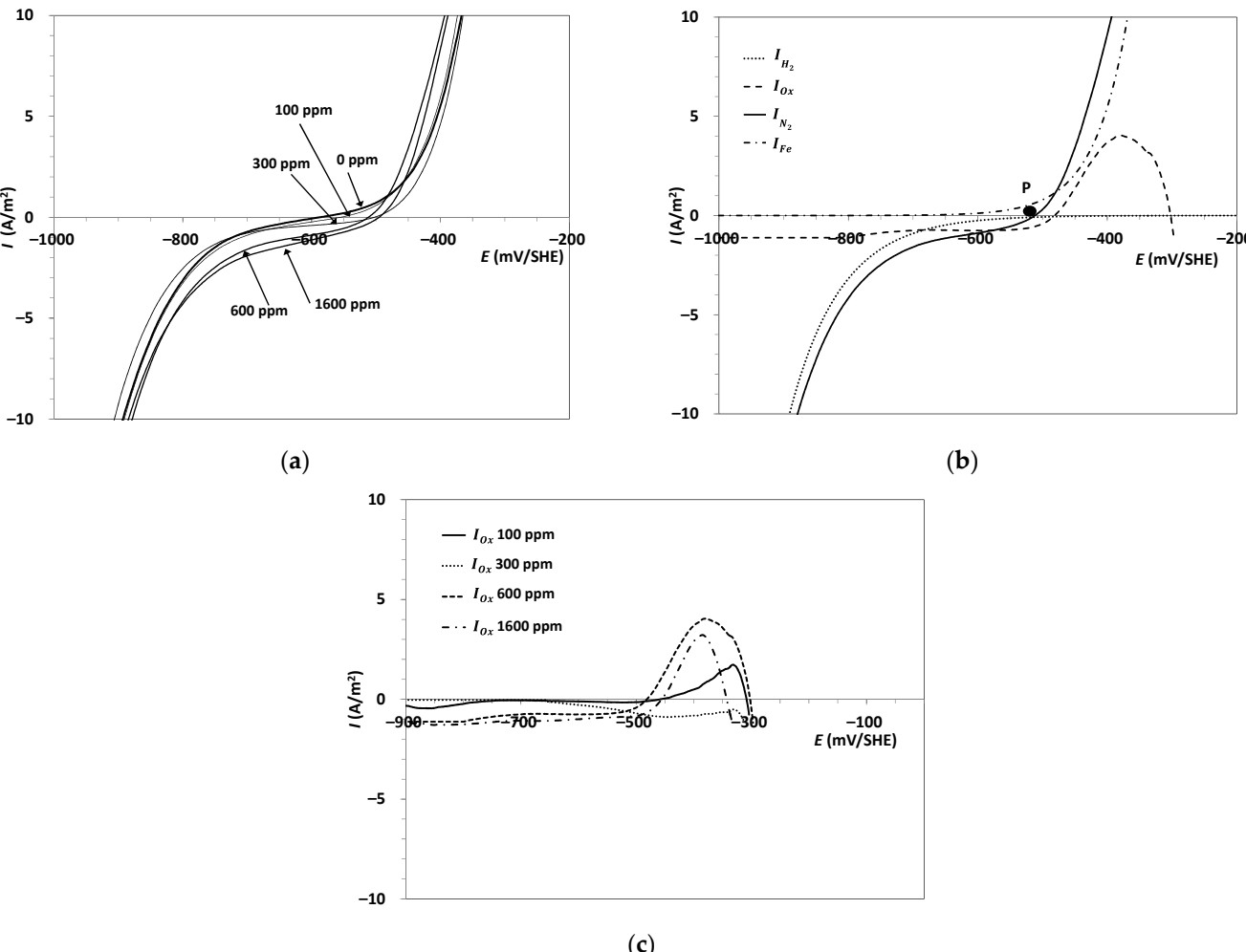

**Figure 4.** (**a**) Early polarization curves for carbon steel in 0.5 M NaCl (N$_2$) at different DR dosing; (**b**) Polarization curves at 600 ppm from Figure 3a and partial curves $I_{H_2}$ (dotted curve), $I_{Fe}$ (dotted-dash curve) and $I_{Ox}$ (dashed curve); and (**c**) $I_{Ox}$ curves at different DR dosing.

Also, it can be assumed that during the early stage of electrode immersion $I_{Fe}^* \approx I_{Fe}$ meaning that the kinetic oxidation of iron is equivalent to that under the absence of DR; under these conditions the partial polarization curves can be estimated as shown in Figure 4b. A family of $I_{Ox}$ is obtained after repeating this process with solutions at increasing doses of DR (Figure 4c). An anodic peak appears in all these curves at about $-350$ mV/SHE from the reduction zone that ends at about $-500$ mV/SHE. The negligible current density for the reduction zone seen at 100 ppm DR suggests that reduction can only takes place after a previous adsorption process of a soluble unknown organic compound in the metal surface that in turn may require a minimum concentration in the bulk phase. Moreover, the corrosion potential value marked as point P in Figure 4b clearly indicates that the concentration of the electroactive component being reduced at the WE surface will decrease gradually with immersion time. Therefore, the $I_{Ox}$ curve will change with time as well as the experimental polarization data.

### 3.3. Corrosion Inhibition under Unaerated Conditions Monitored by LSV and EIS at Different Immersion Times

The corrosion evolution during 24 h of a rotating electrode in a 0.5 M NaCl + 600 ppm DR (N$_2$) was examined by a simultaneous measurement of polarization, EIS, and weight loss using reactor LC as shown in Figure 1b.

The polarization curves for carbon steel at different immersion times (Figure 5a) showed that while $I^*_{H_2}$ remains nearly unaltered, $I_{Ox}$ is displaced upward and the apparent anodic slope changed significantly at immersion times longer than 2 h. A relevant observation was that the solution turned brown after 1 h WE immersion which gradually evolved with time to higher intensity. Remarkably the final electrolyte did not show any trace of precipitated rust even after exposing it to air. After exposure the WE exhibited a dark brown color film firmly attached to the surface.

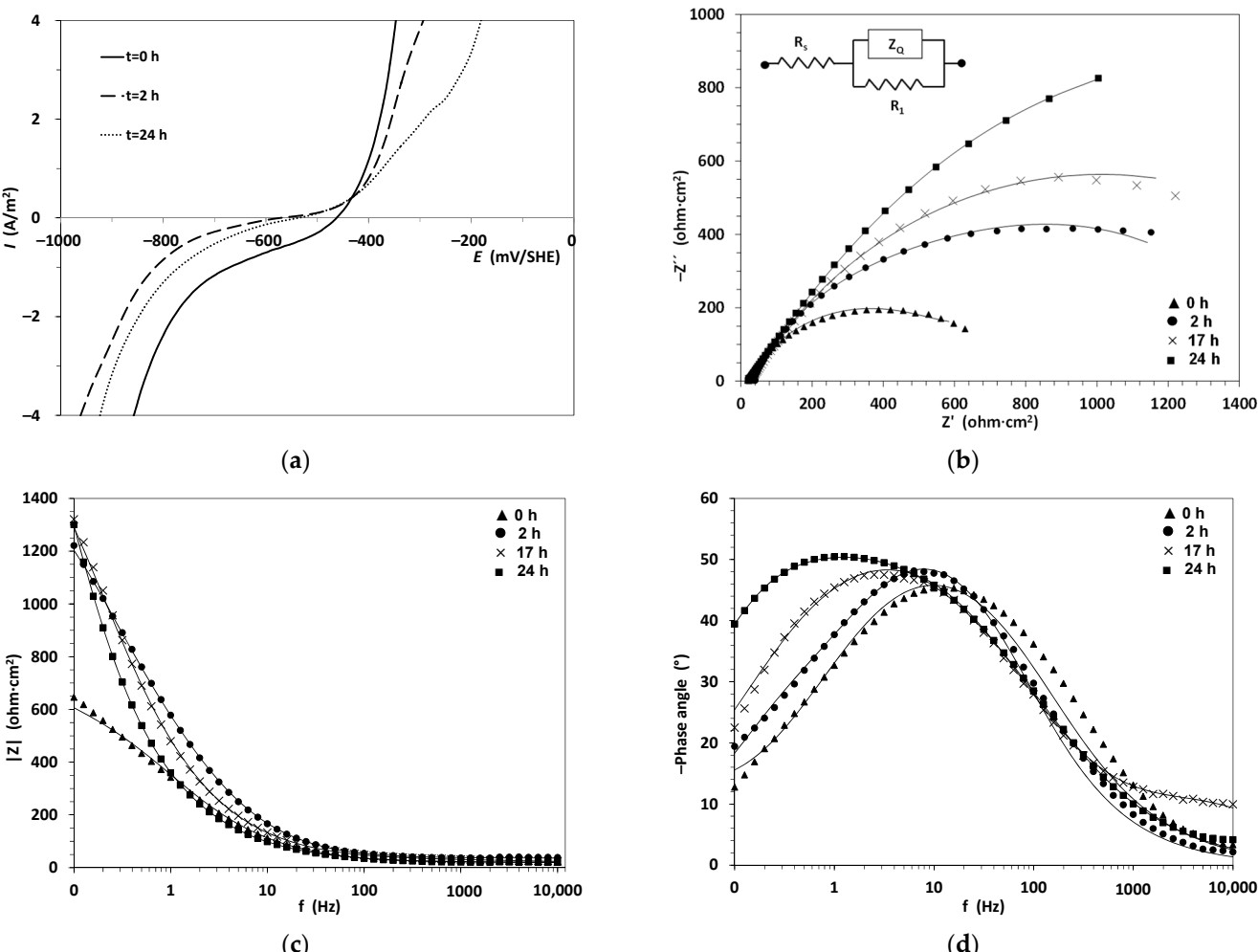

**Figure 5.** Electrochemical results for carbon steel in 0.5 M NaCl + 600 ppm DR (N$_2$) at different immersion times: (**a**) Polarization curves; (**b**) Nyquist diagram; (**c**) Bode magnitude diagram; and (**d**) Bode phase angle diagram. Marks and continuous lines in EIS diagrams indicate the experimental and fitted data, respectively.

Instantaneous $I_{corr}$ values were calculated using the experimental *E-I* data using Equations (A1)–(A7) (see Appendix A) and assuming that the partial oxidation current of any component other than iron, is negligible at the $E_{corr}$ potential. These data show an initial high corrosion rate that gradually decreases until a nearly constant value (Table 2). A consistent global 24 h corrosion rate value determined by the weight-loss method confirms this procedure.

**Table 2.** Corrosion rate (instantaneous and global) of carbon steel in a 0.5 M NaCl + 600 ppm DR ($N_2$) at different immersion times.

| Method | Polarization Curves | | | Weight Loss |
|---|---|---|---|---|
| *t*, h | 0 | 2 | 24 | 24 |
| $I_{corr}$, A/m$^2$ | 0.46 | 0.04 | 0.09 | 0.19 |

Additional evidence was originated from EIS measurements made at time intervals using immersed rotating WE (Figure 5b–d). These data presented as Nyquist plot show a gradual capacity loop evolution toward larger sizes. An appropriate model to fit the experimental data was found to be a resistance element ($R_s$) in series with two parallel elements one resistance ($R_1$) and constant phase element (CPE) defined as:

$$Z_Q = \frac{1}{Q \cdot (w \cdot j)^\alpha} \tag{3}$$

where, $Q$ is a CPE element in s$^\alpha$/$\Omega$·cm$^2$, $\alpha$ is the CPE exponent, and $w$ is the angular frequency in rad/s.

Table 3 shows all the impedance parameters including $R_s$, $R_1$, $Q$, and $\alpha$ fitted from the experimental data using a ZMAN Software of the SP1 Zive Potentiostat. $R_s$ values representing the solution resistance remain almost unchanged with time. The $R_1$ values that steadily increase with time indicate the formation of protective layer probably evolving because of an electrochemical reaction between iron and an unknown DR component.

**Table 3.** Fitted parameter values obtained from EIS data for carbon steel in a 0.5 M NaCl + 600 ppm DR ($N_2$) at different immersion times.

| Immersion Time | EIS Parameters | | | |
|---|---|---|---|---|
| *t*, h | $R_s$, $\Omega$·cm$^2$ | $R_1$, $\Omega$·cm$^2$ | $Q$, s$^\alpha$/$\Omega$·cm$^2$ | $\alpha$ |
| 0 | 18.5 | 743 | $6.5 \times 10^{-4}$ | 0.64 |
| 2 | 35.2 | 1512 | $4.1 \times 10^{-4}$ | 0.67 |
| 17 | 25.1 | 2252 | $6.0 \times 10^{-4}$ | 0.62 |
| 24 | 20.1 | 4180 | $8.5 \times 10^{-4}$ | 0.63 |

Given the slow rate formation of the protective film we can formulate a temporal 24 h inhibition efficiency as:

Using polarization data from Table 2:

$$\eta = \frac{I_{corr}^0 - I_{corr}^{24\,h}}{I_{corr}^0} \times 100 = 80.4\% \tag{4}$$

Using EIS data from Table 3:

$$\eta = \frac{R_1^{24\,h} - R_1^{0\,h}}{R_1^{24\,h}} \times 100 = 82.2\% \tag{5}$$

These can be considered as very good agreements. The theoretical bases for these calculations are found elsewhere [34].

### 3.4. Corrosion Inhibition under Aerated Conditions at Early Immersion Time

Under increasing DR dose, the measured polarization curve moves upward until a limiting dose of 600 ppm, from where the curve keeps almost a fixed position at higher doses (Figure 6a). An important requirement for the validity of the kinetic expressions for the partial electrochemical reactions described above in Section 2.4 is that each curve could be considered as quasi-stationary state. This assumption applies for the present study and

it can be verified as a negligible difference between two consecutive LSV measurements in an electrochemical system. In order to synthetize the partial $I_{O_2}^*$ and $I_{Ox}$ curves the following assumptions were made: (a) the oxygen reduction reaction is precluded by DR adsorption on the metal surface in such a way that only the limiting current density $I_l$ is altered. The change in $I_l$ between the curves at 0 and 1600 ppm DR is determined at $-700$ mV (Figure 6a) as $\Delta I_l \approx 1.5$ A/m$^2$. This is because the parallel curves at potentials more negative than $-700$ mV/SHE suggest predominance of electrochemical reactions for $O_2$ and $H_2$; (b) for the same reason mentioned in the preceded section both the partial hydrogen evolution and iron oxidation reactions are not altered in the presence of DR, so that $I_{H_2}^* = I_{H_2}$ and $I_{Fe}^* = I_{Fe}$. The partial $I_{O_2}^*$ curve using equation 5 at different DR dose were calculated from $I_l^*$ values proportionally distributed, as shown in Table 4, and the $a_{O_2}$ and $t_{cO_2}$ values from Table 1. Subsequently, the $I_{Ox}$ curves were determined as $I_{Ox} = I^* - I_{O_2}^* - I_{H_2} - I_{Fe}$. The peak seen in Figure 6b is very similar to that on Figure 4c and shows a small distortion at about $-370$ mV/SHE which can be originated from changes in parameters $a_{O_2}$ or $t_{cO_2}$ at different DR doses.

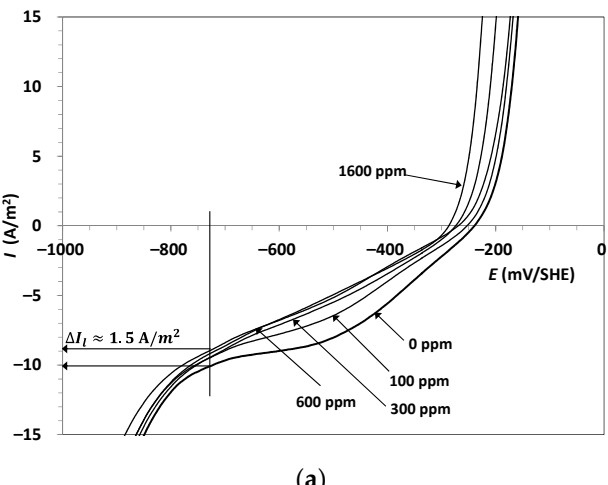
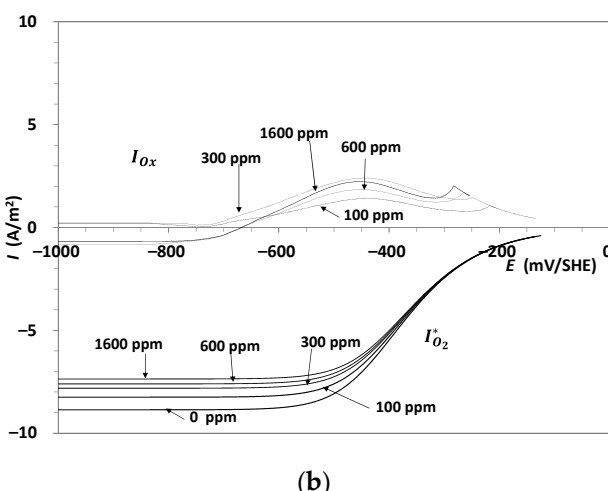

(**a**)                                    (**b**)

**Figure 6.** (**a**) Early LSV measurements for carbon steel in 0.5 M NaCl (air) at different DR dosing; and (**b**) Synthetized $I_{ox}$ curves at different DR dosing from data (**a**) of this figure.

**Table 4.** Estimated $I_l^*$ values in 0.5 M NaCl (air) at initial immersion time and different DR dose.

| DR Dosing, ppm | $I_l^*$, A/m$^2$ |
|:---:|:---:|
| 0 | $-8.9$ |
| 100 | $-8.3$ |
| 300 | $-7.8$ |
| 600 | $-7.6$ |
| 1600 | $-7.4$ |

Tentative numerical simulations with different $a_{O_2}$ and $t_{cO_2}$ values show that this distortion can be minimized while the peak is still in place.

### 3.5. Corrosion Inhibition under Aerated Condition Monitored with LSV and EIS at Different Immersion Times

The corrosion evolution of a rotating electrode at 600 and 1200 ppm DR, in 0.5 M NaCl under aerated condition, was examined for up to 48 h with simultaneous measurements of polarization, EIS, and weight loss in an LC reactor. Polarization and EIS results are shown in Figures 7 and 8, respectively.

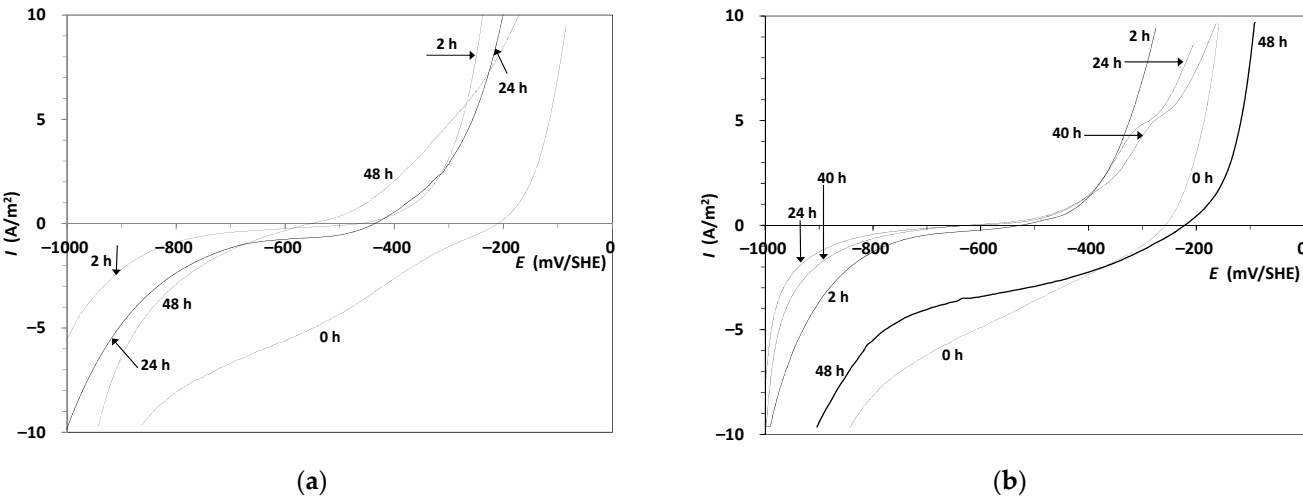

**Figure 7.** Polarization curves for carbon steel in aerated conditions at different immersion times: (**a**) Polarization curves considering 600 ppm DR (air); and (**b**) Polarization curves considering 1200 ppm DR (air).

**Figure 8.** *Cont.*

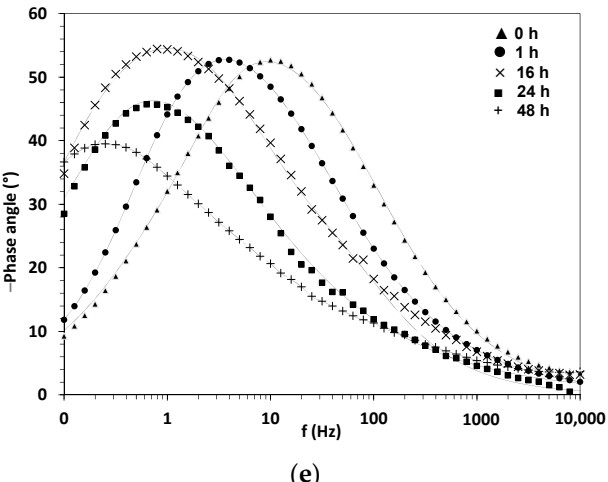
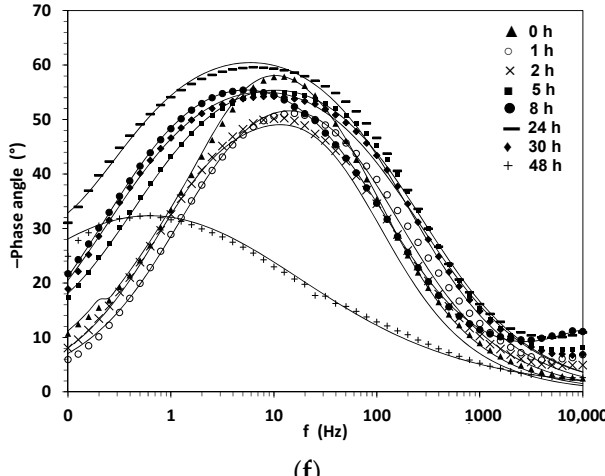

(**e**)          (**f**)

**Figure 8.** Impedance measurements for carbon steel in aerated conditions at different immersion times: (**a**) Nyquist diagram for 600 ppm DR (air) (**b**) Nyquist diagram for 1200 ppm DR (air); (**c**) Bode magnitude diagram for 600 ppm DR (air); (**d**) Bode magnitude diagram for 1200 ppm DR (air); (**e**) Bode phase angle diagram for 600 ppm DR (air); and (**f**) Bode phase angle diagram for 1200 ppm DR (air). Marks and continuous lines in EIS diagrams indicate the experimental and fitted data, respectively.

Polarization curves for carbon steel at different immersion times (Figure 7) show that while $I_{H_2}^*$ remains nearly unaltered, the absolute values of the partial current densities for both $I_{O_2}^*$ and $I_{Ox}$ gradually decrease so that the instantaneous $I_{corr}$ values also decrease producing an overall *IE* efficiency of 68% and 87% (from weight-loss measurements) over the time span of 48 h for 600 and 1200 ppm DR and such as is tabulated in Tables 5 and 6, respectively. It is interesting to note that the $I_{corr}$ values at early immersion time are like those observed at a plain 0.5 M NaCl solution. This suggests an early stage of iron corrosion followed by a complex formation between $Fe^{+2}$ or $Fe^{+3}$ with some of the constituents identified for the *Skytantus acutus* (see Section 3.6) that, subsequently, is adsorbed in the metal surface. The color evolution and the absence of solid residue for corrosion runs under $N_2$ were also seen under air. These observations can also be considered as an indication of such complex formation [35].

**Table 5.** Corrosion rate (instantaneous and global) of carbon steel in a 0.5 M NaCl (air) + 600 ppm DR at different immersion times.

| Method | Polarization Curves | | | | Weight Loss With DR | Weight Loss Without DR | *IE* % |
|---|---|---|---|---|---|---|---|
| *t*, h | 0 | 2 | 24 | 48 | 48 | 48 | - |
| $I_{corr}$, A/m² | 0.91 | 0.22 | 0.37 | 0.23 | 0.41 | 1.26 | 68 |

**Table 6.** Corrosion rate (instantaneous and global) of carbon steel in a 0.5 M NaCl (air) + 1200 ppm DR at different immersion times.

| Method | Polarization Curves | | | | | Weight Loss With DR | Weight Loss Without DR | *IE* % |
|---|---|---|---|---|---|---|---|---|
| *t*, h | 0 | 2 | 24 | 40 | 48 | 48 | 48 | - |
| $I_{corr}$, A/m² | 1.17 | 0.16 | 0.06 | 0.11 | 0.97 | 0.17 | 1.26 | 87 |

Forty one per cent of the dissolved iron remained as part of the film attached on the metal surface during 24 h corrosion run with 600 ppm DR. This iron mass balance considered the total dissolved iron by corrosion and the iron concentration in solution measured by flame atomic spectroscopy analysis. Interestingly, there was no traces of solid residues in solution, as it was observed in runs under $N_2$. This observation is indicative of

the absence of oxidation of ferrous iron to ferric hydroxide and subsequent precipitation as rust in liquid phase in the presence of DR.

According to EIS measurements at increasing immersion times as shown in Nyquist plots (Figure 8a,b), the capacity loop size increased with time up to a maximum intermediate value that further decreases to a final size lower than the earliest loop. An appropriate model to fit the experimental data was found to be the same former circuit (insert in Figure 8) in series with additional two parallel elements, one resistance, and a constant phase element. The parameter $R_s$ values (Tables 7 and 8) representing the solution resistance, have the same variation range that those seen in Table 3. The $R_T$ ($R_T = R_1 + R_2$) parameter follows a similar variational trend as $I_{corr}$ values seen in Tables 5 and 6. All other parameters that seem to change around a constant mean do not follow a distinctive trend with time. Our interpretation for this EIS information is the evolution of a film resulting from iron oxidation, precipitation, and adhesion on the surface and gradual detachment, which is probably due to shear stress exerted from the steel specimen rotating at 1200 rpm. A gradual film degradation (or detachment) with time was also observed in a former work [36] focused on the influence of hydrodynamic conditions on the behavior of an inhibitor film in a rotating disk electrode.

**Table 7.** Fitted parameter values obtained from EIS data for carbon steel in a 0.5 M NaCl (air) + 600 ppm DR at different immersion times.

| Immersion Time | EIS Parameters | | | | | | | |
|---|---|---|---|---|---|---|---|---|
| $t$, h | $R_s$, $\Omega \cdot cm^2$ | $R_1$, $\Omega \cdot cm^2$ | $Q_1$, $s^\alpha/\Omega \cdot cm^2$ | $\alpha_1$ | $R_2$, $\Omega \cdot cm^2$ | $Q_2$, $s^\alpha/\Omega \cdot cm^2$ | $\alpha_2$ | $R_T$, $\Omega \cdot cm^2$ |
| 0 | 12.1 | 64.4 | $1.1 \times 10^{-2}$ | 0.98 | 349.8 | $5.6 \times 10^{-4}$ | 0.78 | 414.3 |
| 1 | 20.3 | 186.1 | $3.1 \times 10^{-3}$ | 0.59 | 491.9 | $6.7 \times 10^{-4}$ | 0.89 | 678.0 |
| 16 | 15.4 | 249.2 | $4.6 \times 10^{-3}$ | 0.61 | 624.8 | $2.0 \times 10^{-3}$ | 0.91 | 874.0 |
| 24 | 18.9 | 167.9 | $4.0 \times 10^{-3}$ | 0.59 | 277.3 | $5.2 \times 10^{-3}$ | 0.95 | 445.3 |
| 48 | 10.8 | 189.4 | $1.3 \times 10^{-2}$ | 0.80 | 237.3 | $2.1 \times 10^{-2}$ | 0.34 | 426.7 |

**Table 8.** Fitted parameter values obtained from EIS data for carbon steel in a 0.5 M NaCl + 1200 ppm DR (air) at different immersion times.

| Immersion Time | EIS Parameters | | | | | | | |
|---|---|---|---|---|---|---|---|---|
| $t$, h | $R_s$, $\Omega \cdot cm^2$ | $R_1$, $\Omega \cdot cm^2$ | $Q_1$, $s^\alpha/\Omega \cdot cm^2$ | $\alpha_1$ | $R_2$, $\Omega \cdot cm^2$ | $Q_2$, $s^\alpha/\Omega \cdot cm^2$ | $\alpha_2$ | $R_T$, $\Omega \cdot cm^2$ |
| 0 | 9.5 | 90 | $8.1 \times 10^{-4}$ | 0.99 | 388 | $1.2 \times 10^{-3}$ | 0.72 | 478 |
| 1 | 14.2 | 37 | $7.6 \times 10^{-4}$ | 0.75 | 453 | $3.9 \times 10^{-4}$ | 0.84 | 490 |
| 2 | 24.0 | 166 | $5.9 \times 10^{-4}$ | 0.8 | 445 | $6.6 \times 10^{-4}$ | 0.82 | 611 |
| 5 | 10.0 | 200 | $1.7 \times 10^{-3}$ | 0.74 | 951 | $6.7 \times 10^{-4}$ | 0.71 | 1151 |
| 8 | 18.5 | 19 | $9.5 \times 10^{-2}$ | 1 | 1794 | $4.2 \times 10^{-4}$ | 0.72 | 1813 |
| 24 | 13.9 | 92 | $8.3 \times 10^{-4}$ | 0.81 | 2953 | $3.8 \times 10^{-4}$ | 0.78 | 3045 |
| 30 | 12.6 | 120 | $7.7 \times 10^{-3}$ | 1 | 1466 | $4.5 \times 10^{-4}$ | 0.71 | 1586 |
| 48 | 19.7 | 85 | $1.6 \times 10^{-2}$ | 0.99 | 241 | $4.9 \times 10^{-3}$ | 0.49 | 327 |

The fixed equivalent circuit with time also suggests that the film composition and structure is time independent and on a set of electrochemical reactions that generates the protective film triggered by unknown component in the DR. This component that gradually decreases in concentration would require periodic DR addition to maintain the inhibiting action for longer times. In industrial practice corrosion inhibitors are continuously dosed in flowing piping systems so that a constant concentration of the active component is maintained; significant corrosion rate variability detected by on-line corrosion rate sensors takes place normally only at seasonal periods [37] so that the concentration of the active component should not be a restrictive factor.

### 3.6. Corrosion Rate Density and Inhibition Efficiency Data for 24 h Runs at Three Different Rotation Rates and DR Dose

Data on Table 9 for $I_{corr}$ and *IE*, measured by weight loss for corrosion runs not monitored during immersion, are similar to those presented in former sections. This is an indication that electrochemical measurements taken during 24 h immersion time do not disrupt significantly the protecting film on the metal surface.

**Table 9.** $I_{corr}$ and *IE* data for carbon steel in a 0.5 M NaCl with 24 h immersion at different DR dose.

| Rotation Rate: 0 rpm | | | | | |
|---|---|---|---|---|---|
| **DR dosing, ppm** | 0 | 100 | 300 | 600 | 1600 |
| $I_{corr}$, **A/m$^2$** | 0.419 | 0.170 | 0.154 | 0.047 | 0.056 |
| *IE*, % | - | 59.3 | 63.2 | 88.7 | 86.7 |
| **Rotation Rate: 1200 rpm** | | | | | |
| **DR dosing, ppm** | 0 | 100 | 300 | 600 | 1600 |
| $I_{corr}$, **A/m$^2$** | 1.260 | 0.972 | 0.572 | 0.248 | 0.294 |
| *IE*, % | - | 22.8 | 54.6 | 80.3 | 76.7 |
| **Rotation Rate: 3000 rpm** | | | | | |
| **DR dosing, ppm** | 0 | 100 | 300 | 600 | 1600 |
| $I_{corr}$, **A/m$^2$** | 2.091 | 2.007 | 0.550 | 0.410 | 0.229 |
| *IE*, % | - | 4.0 | 73.7 | 80.4 | 89.4 |

A rust residue at the bottom of the cell was only observed up to a dose of 300 ppm DR; at larger doses and although dark-colored, the liquid was translucid without a trace of suspended solid. This observation is consistent with IE data lower than 70% which is associated to a minimum DR dose required to either fully cover the metal surface and/or fulfill a complete reaction with iron ions to form an insoluble film from a precipitated iron complex.

For the rotation rate between 0 to 3000 rpm and at a dose larger than 600 ppm DR does not influence the *IE*; then this dose can be considered optimal (Figure 9). In the case of 100 ppm DR, the sharp drop in *IE* seen at increasing and sufficiently large rotation rates would be a consequence of film removal from covered and susceptibly isolates surface sections.

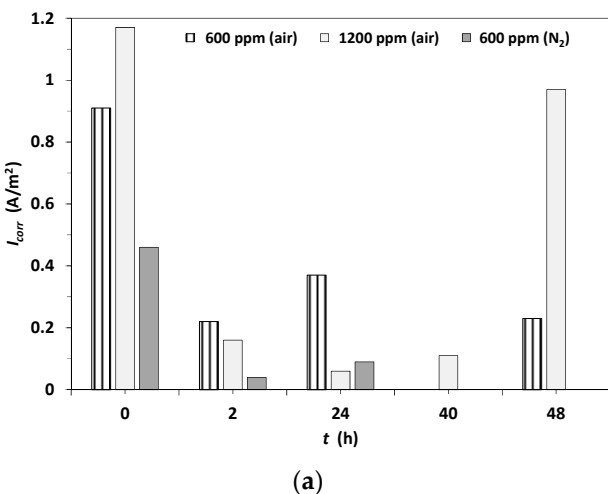 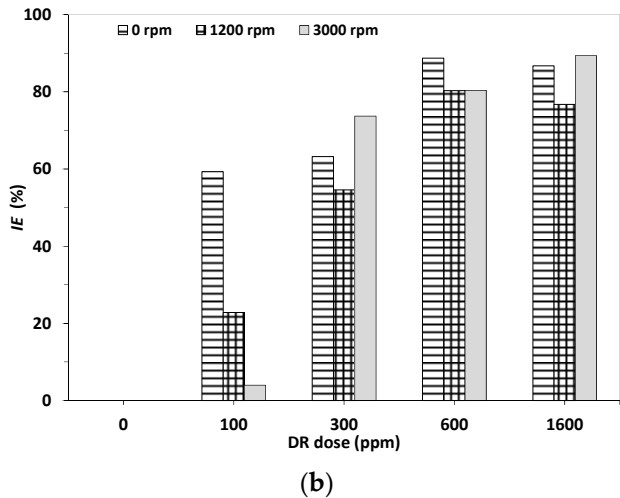

(a)  (b)

**Figure 9.** (**a**) Corrosion rate evolution for carbon steel at 1200 ppm under air and N$_2$ bubbling; and (**b**) *IE* after 24 h immersion at different DR dose and rotation rates in air bubbling.

### 3.7. Corrosion Rate Density Evolution with Time: Global Comparison

3.7.1. Corrosion at Early Immersion Time

A comparison between the different $I_{corr}$ values that are shown in Figure 9a shows that the early $I_{corr}$ value under $N_2$ without DR is significantly lower than that with 600 ppm DR dose; this indicates that a DR component exacerbate the early iron corrosion in the absence of oxygen. Given that dissolved iron is a requirement for a complex formation [38,39] the early iron corrosion mechanism could be explained in terms of catalytic action exerted by a DR component. On the other hand, in the presence of air $I_{corr}$ values without DR are lower than those with DR; this suggests that a fast iron complexing kinetic with subsequent adsorption takes place in the presence of the dissolved iron from the initial corrosion promoted by dissolved oxygen.

3.7.2. Corrosion Evolution with Different DR Dose

The largest inhibition efficiency value observed at 100 ppm DR dose is under quiescent conditions (0 rpm) (Figure 9b) where the rotation rate is relevant. In contrast, at larger DR dose the incidence of rotation is not a relevant factor. It suggests that at 100 ppm the carbon steel surface is partially covered by an adsorbed organic layer and there are no iron ions available for complexing because the corrosion process is diverted just to the uncovered surface fraction; therefore, the organic layer in the covered fraction exerts an efficient corrosion protection. WE rotation rates between 0 and 3000 rpm seem to be irrelevant at larger DR doses probably because the complete steel surface is covered by an adsorbed layer. This also suggests that the adsorbed organic layer reacts with iron ions generated by corrosion only after the surface coverage is complete. Along with this reaction the $I_{corr}$ values could decrease up to a minimum value where the complexing reaction is complete. A further $I_{corr}$ increase from this minimum value would be a result from a slow layer degradation and/or detachment. Thus, the integrity of the organic layer as shield to mitigate corrosion would require a permanent supply of inhibitor to maintain low $I_{corr}$ values during long periods of time.

### 3.8. Dried Extract Characterization

The main identified DR constituents from *Skytanthus acutus* were phenolic compounds (Figure 10) usually found in some plant tissues. Two interesting features of these compounds are: (a) carbon steel corrosion inhibiting activity in HCl [40,41] and chloride solutions [42,43] have been reported in several plant extracts containing quinin derivatives, ascorbate, gallocatechin, protocatechuic acid and chlorogenic acid; and (b) all these types of compounds are partially water soluble, and their relative abundance should be proportional to its solubility. Quinic acid showed the highest peak signal (see peak 1 in Figure 10) and a high water solubility [44]; protocatechuic acid, chlorogenic acid, and patuletin (peaks 6, 9–10, 16 in Figure 10) have low to moderate solubilities [45].

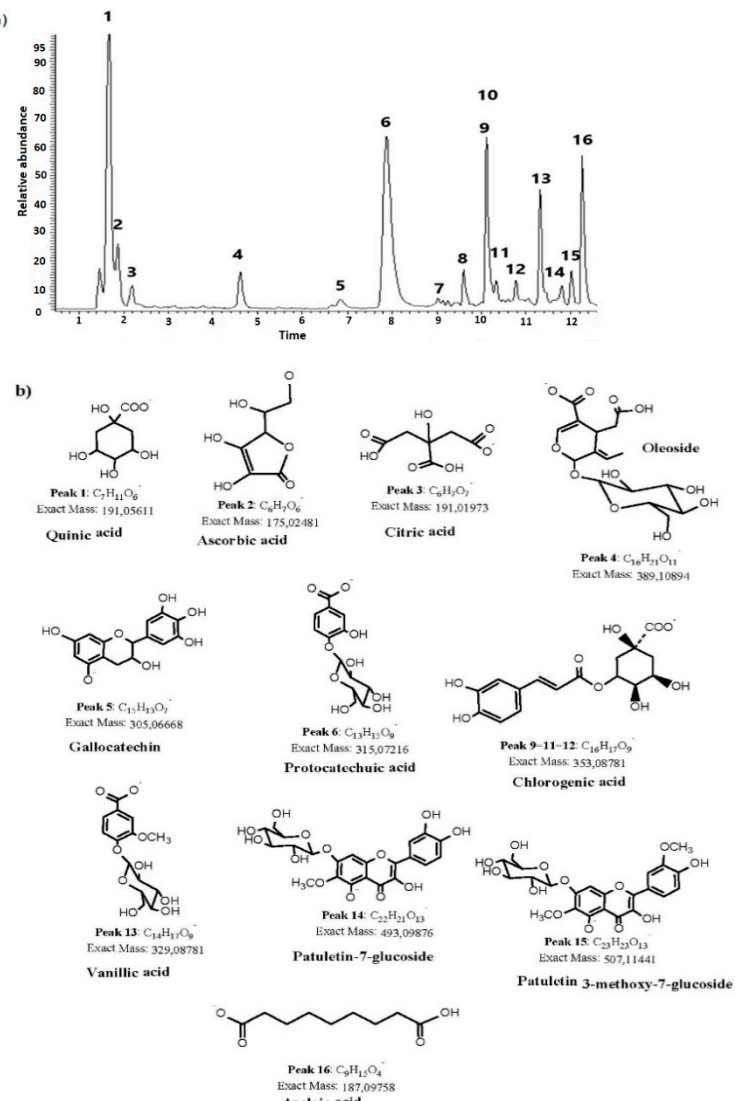

**Figure 10.** (**a**) UHPLC-TIC (ultra HPLC total ion current) of compounds identified in DR of *Skytanthus acutus M.*; and (**b**) Structure for each peak number.

### 3.9. Corrosion Inhibition Mechanism

Iron and steel surfaces are covered with a thin oxide layer soon after air exposure. Metallic iron is thermodynamically unstable in the presence of water and corrosion proceeds with $Fe^{2+}$ ions generated by electrochemical mechanisms; this process is exacerbated by the presence of $Cl^-$ ions [46,47]. On the other hand, a variety of organic compounds, polyphenols among them, have the ability to bind to Fe [35,48]. Dissolved ferrous iron can be maintained in aquatic systems containing organic species of natural origin [49,50]. Also, phenolic reductants widely distributed in plants have been used to dissolve iron minerals such as goethite and hematite [51,52]. Reactions of catechol, chlorogenic, and protocatechuic acids with iron involve complex formation with $Fe(III)$ or $Fe(II)$ in aqueous media [53]; and under certain conditions a subsequent decomposition of protecatechoic-$Fe(III)$ complexes can takes place [54]. Catechin can be oxidized to form quinones while $Fe(III)$ is reduced to $Fe(II)$ [38]. The corrosion inhibition of carbon steel in chloride media by ascorbic acid has been attributed to its ability to form chelates of various solubilities and different oxidation states [42,55]. Also, vanillic acid prevents $Fe(II)$ oxidation [56]. Iron complexes formed by phenolics and $Fe(III)$ complexes have been reported, and in many cases dark colored precipitated products have been observed immediately after reaction [35,39,57]. The iron binding constant of many phenolic compounds, including

protocatechuic acid and chlorogenic acid, have been compared with those from EDTA, but with lower complexing abilities than EDTA, a well-known chelator [58].

In order to propose a mechanism for the corrosion inhibition of carbon steel probes under high saline conditions, mediated by dried extracts from *Skytanthus acutus* leaves, the present research provides the following support:

- $I_{corr}$ values for carbon steel in 0.5 M NaCl are similar to those obtained in the presence of the inhibitor. Also, $I_{corr}$ gradually decrease with the immersion time;
- Partial polarization curves emerge from a mixed potential analysis that gradually evolve with time. The synthesized $I_{Ox}$ curves indicate that some components (shown in Figure 10) are electroactive at the potential range that carbon steel corrosion takes place;
- Forty one per cent of the dissolved iron from the corrosion process remained as film attached on the metal surface during 24 h corrosion runs with 600 ppm DR;
- At DR doses larger than 300 ppm, the dark color-evolving solutions remained translucent. Absence of precipitates was observed in solutions left to rest at room temperature for several months. Color formation together with corrosion inhibition takes place irrespective of the presence of if dissolved oxygen in solution. Thus, electroactive components in DR can either be oxidized or reduced at the expense of $Fe(III)$ reduction or $Fe(II)$ oxidation; and
- Combined EIS and LSV measurements demonstrated that film formation evolves with time, provides an increasing corrosion protection of carbon steel in chloride media, and after a limiting time when the film gradually degrades with a subsequent increase in $I_{corr}$ values. This would suggest that after a phenolics-$Fe(III)$ complex is formed a subsequent decomposition or detachment would take place.

Thus, we propose the following mechanism involving four different processes taking place simultaneously, and compatible with the above-mentioned evidence: first, the adsorption of organic ligands from the solution to the carbon steel surface. It is important to stress that this process should proceed in different ways depending on the nature of each DR constituent [59]; second, the iron dissolution process should be initiated by an induction stage by which the generation of ferrous ions in the solution that takes place under a water corrosion mechanism exacerbated by Cl$^-$ ions, followed by a catalytic stage induced by adsorbed organic constituents; third, a complex formation between ferric and ferrous with DR constituents along with a complicated array of global stability constants and physico-chemical properties; and, fourth, the complex desorption where a fraction of the complex formed in the surface are released into solution. This process could have a time-dependent evolution.

Finally, in a global perspective, the temporal film evolution seems to be formed by two opposing processes consisting of an iron complex formation on the metal surface, followed by film detachment, desorption, or degradation. The protecting film, using the dried residue dosed in a batch mode, will be gradually degraded as the concentration of the active component for the iron complex decreases with time.

## 4. Conclusions

The experimental evidence reported here allows the following conclusions:

- Dry aqueous extracts from *Skytanthus acutus* leaves, a native plant from the Atacama Desert in northern Chile, are active inhibitors of carbon steel corrosion in 0.5 M NaCl;
- The dry extract inhibitor conveys easy handling, stability, and storage for long period of time without degradation;
- From the mixed potential theory applied to polarization *E-I* data for carbon steel in NaCl solutions dosed with DR, the total current can be expressed as a superposition of partial current for hydrogen evolution, oxygen reduction, iron oxidation, and global redox reaction from component(s) present at the *Skytanthus acutus* extract;

- Based on the mixed potential theory it was demonstrated that a redox activity of the *Skytanthus acutus* dry extract components is involved in the corrosion inhibition mechanism of carbon steel in 0.5 M NaCl;
- The corrosion inhibition efficiency of doses larger than 300 ppm of the dry extract is not significantly affected by a rotation rate as high as 3000 rpm;
- The optimum DR dose for carbon steel inhibition is 600 ppm with a maximum inhibition efficiency of 90%;
- The corrosion inhibition efficiency gradually decreases with time when DR is added on a batch basis; and
- The experimental evidence indicated that the inhibition mechanisms can be formulated in terms of four stages of adsorption of organic ligands, iron dissolution, complex formation, and complex desorption.

**Author Contributions:** Conceptualization: L.C., B.G.-S. and A.S.; methodology: L.C. and A.S.; software and validation: L.C. and F.G.; Formal analysis: B.G.-S. and L.C.; investigation: Y.F. and F.G.; resources: B.G.-S., L.C. and J.B.; metabolite characterization: J.B.; data curation: F.G. and A.S.; writing—original draft preparation: L.C. and B.G.-S.; writing—review and editing: A.S. and J.B.; supervision: L.C.; funding acquisition: B.G.-S. All authors have read and agreed to the published version of the manuscript.

**Funding:** This research was funded by CONICYT Chile, grant number CeBiB FB-0001.

**Institutional Review Board Statement:** Not applicable.

**Informed Consent Statement:** Not applicable.

**Data Availability Statement:** The processed data are available from the corresponding author upon request.

**Conflicts of Interest:** The authors declare no conflict of interest.

## Nomenclature

| | |
|---|---|
| AISI 1020 | Steel low-tensile carbon steel |
| ASTM D2688–05 | Standard Test Methods for Corrosivity of Water in the Absence of Heat Transfer (Weight-Loss Methods) |
| ASTM G1-90 | Standard Practice for Preparing, Cleaning, and Evaluating Corrosion Test Specimens |
| HPLC-MS | Liquid Chromatography–Mass Spectrometry |
| UHPLC-PDA-HESI-orbitrap-MS/MS | Ultra-High-Performance Liquid Chromatography–Tandem Mass Spectrometry and a Photodiode Array Detector |
| UHPLC | Ultra-High-Performance Liquid Chromatography |
| EIS | Electrochemical Impedance Spectroscopy |
| DR | Dried Residue |
| MS | Mass Spectrometry |
| AgCl | Silver Chloride |
| KCl | Potassium Chloride |
| NaCl | Sodium Chloride |
| HCl | Hydrochloric acid |
| Ag | Silver |
| C | Carbon |
| Mn | Manganese |
| Fe | Iron |
| S | Sulfur |
| Ni | Nickel |
| Cr | Chromium |
| Sn | Tin |
| P | Phosphorus |
| Mo | Molybdenum |
| $N_2$ | Nitrogen gas |

| SC | Small Cell |
|---|---|
| LC | Large Cell |
| LSV | Linear Sweep Voltammetry |
| SHE | Standard Hydrogen Electrode |
| RPM | Revolutions per minute |
| PTFE | Polytetrafluoroethylene |
| WE | Working Electrode |
| HE | Hydrogen Evolution |
| ORR | Dissolved Oxygen Reduction |
| IO | Iron Oxidation |
| E | Potential |
| $I$ | Total current density |
| $I_{O_2}$ | Partial Reduction Current for Dissolved Oxygen Reduction |
| $I_{H_2}$ | Partial reduction Current for Hydrogen Evolution |
| $I_{0,O_2}$ | Exchange Current Density for Dissolved Oxygen Reduction |
| $I_{0,H_2}$ | Exchange Current Density for Hydrogen Evolution |
| $I_{0,Fe}$ | Exchange Current Densities for Iron Oxidation |
| $I_l$ | Limiting Current Density |
| $\Delta I_l$ | Delta Limiting Current Density |
| $I^*$ | Total Current Density in Presence of Inhibitor |
| $I^*_{O_2}$ | Exchange Current Density in Presence of Inhibitor for Dissolved Oxygen Reduction |
| $I^*_{H_2}$ | Exchange Current Density in Presence of Inhibitor for Hydrogen Evolution |
| $I^*_{Fe}$ | Exchange Current Density in Presence of Inhibitor for Iron Oxidation |
| $I_{Ox}$ | Exchange Current Density in Presence of Inhibitor for Global Redox Reaction (plant-extract components) |
| IE | Inhibition Efficiency |
| $I_{corr}$ | Corrosion Rates Density |
| $I^o_{corr}$ | Corrosion Rates Density in Absence of the Inhibitor |
| $I^i_{corr}$ | Corrosion Rates Density in Presence of the Inhibitor |
| $\eta_{O_2}$ | Overpotential for Dissolved Oxygen Reduction |

## Appendix A

Kinetic expressions and their simplifications for ORR, HE, and IO partial reactions considering pure charge transfer control and a mixed-mass charge transfer and diffusion mechanism.

$$I = I_{O_2} + I_{H_2} + I_{Fe} \tag{A1}$$

$$I_{O_2} = I_{0,O_2} \frac{e^{-2.303(\frac{\eta_{O_2}}{t_{cO2}})}}{2I_l} \left[ -I_{0,O_2} e^{-2.303(\frac{\eta_{O_2}}{t_{cO2}})} + \sqrt{\left(I_{0,O_2} e^{-2.303(\frac{\eta_{O_2}}{t_{cO2}})}\right)^2 + 4I_l^2} \right] \tag{A2}$$

$$I_{H_2} = I_{0,H_2} e^{-2.303(\frac{\eta_{H_2}}{t_{cH2}})} \tag{A3}$$

$$I_{Fe} = I_{0,Fe} e^{-2.303(\frac{\eta_{Fe}}{t_{aFe}})} \tag{A4}$$

$$I_{O_2} = \frac{A}{2I_l} \left[ -A + \sqrt{A^2 + 4I_l^2} \right] \text{ with } A = a_{O_2} exp\left( -\frac{2.303E}{t_{cO_2}} \right) \tag{A5}$$

$$I_{H_2} = a_{H_2} exp\left( -\frac{2.303E}{t_{cH_2}} \right) \tag{A6}$$

$$I_{Fe} = a_{Fe} exp\left( \frac{2.303E}{t_{aFe}} \right) \tag{A7}$$

$$I_{0,O_2} = a_{O_2} exp\left( \frac{2.303E_{eqO_2}}{t_{cO_2}} \right) \tag{A8}$$

$$I_{0,H_2} = a_{H_2} exp\left(\frac{2.303 E_{eqH_2}}{t_{cO_2}}\right) \tag{A9}$$

$$I_{0,Fe} = a_{Fe} exp\left(-\frac{2.303 E_{eqFe}}{t_{aFe}}\right) \tag{A10}$$

where, $a_{O_2}$, $a_{H_2}$ and $a_{Fe}$ are constants for ORR, HE, and IO, respectively, obtained from the *E-I* polarization curves fitting.

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
