# Peer review of "Aqueous Dried Extract of Skytanthus acutus Meyen as Corrosion Inhibitor of Carbon Steel in Neutral Chloride Solutions"

_metals, doi:10.3390/met11121992_

Round 1
Reviewer 1 Report
Can the use of inhibitor allow the use of lower grade carbon steel, which significantly reduces the capital costs of a project compared to the use of higher grade alloys in the same project?
Corrosion inhibitors are effective only for a particular metallic material in a particular environment. Minor changes in solution composition ( Solvent )can significantly alter the inhibition efficacy,
How did you come to determine that percentage?
The upper temperature limit is one of the most important parameters in the selection of the inhibitor, since some components are sensitive to thermal decomposition, i.e., they lose inhibition effectiveness.
are inhibitors effective for ferrous materials at different temperature levels?
Author Response
We thank you for giving us the opportunity to revise our manuscript “Aqueous dried extract of Skytanthus acutus as corrosion inhibitor of carbon steel in neutral chloride solutions”, for publication in Metals journal. We want to express our appreciation for taking the time and effort necessary to provide an insightful guidance. We hope that these revisions improve the paper such that you now deem it worthy of publication in Metals journal.
Reviewer 1
1.- Can the use of inhibitor allow the use of lower grade carbon steel, which significantly reduces the capital costs of a project compared to the use of higher grade alloys in the same project?
A sentence to cover this issue have been added in the introduction:
“Although carbon steel has no capacity to develop a protective surface oxide layer against the aggressiveness of the surrounding environment, it is the most widely used steel material in engineering applications. This is justified by its significant lower cost in comparison to more noble higher-grade alloys, mechanical properties, and its amenability to be safely operated in very corrosive service conditions under properly designed and implemented control system. In such cases corrosion inhibitors are added in very small amounts that adhere to the metal surface to form a protective barrier against corrosive agents contacting the metal. The efficiency of an inhibitor to provide corrosion protection depends to a large extent upon the interactions between the inhibitor and the metal surface [1]”.
Corrosion inhibitors are effective only for a particular metallic material in a particular environment. Minor changes in solution composition (Solvent)can significantly alter the inhibition efficacy,
How did you come to determine that percentage?
Once an inhibitor formulation has been developed for a particular application (for example: inhibitor for carbon steels in neutral chloride media), then the optimum dose is determined by trial-and-error testing (either electrochemical and/or weight loss standard methods) using increasing inhibitor concentrations not exceeding 1 g/L. In the industrial practice these tests are periodically repeated to cope with a variability in composition of the corrosive media.
The upper temperature limit is one of the most important parameters in the selection of the inhibitor, since some components are sensitive to thermal decomposition, i.e., they lose inhibition effectiveness.
are inhibitors effective for ferrous materials at different temperature levels?
Yes, some inhibitors formulations have been developed for high temperature and pressure. For example there are some deep wells operating at temperatures above 150°C that use corrosion inhibitors for protection. These are rare cases.
In the range below 100 °C inhibitor formulations are available for different temperature ranges. For the case of corrosion inhibition of carbon steel immersed in seawater normally corrosion inhibitors are equally effective in all natural temperature ranges.
Reviewer 2 Report
Thank you for submitting your paper. The work done here draws attention to a significant subject in characterisation of carbon steel. I have found the paper to be interesting. However, several issues need to be addressed properly before the paper is being considered for publication. My comments including major and minor concerns are given below:
- Please consider reviewing the abstract and highlight the novelty, major findings, and conclusions. I suggest reorganizing the abstract, highlighting the novelties introduced. The abstract should contain answers to the following questions:
- What problem was studied and why is it important?
- What methods were used?
- What conclusions can be drawn from the results? (Please provide specific results and not generic ones). Please use numbers or % terms to clearly shows us the results in your experimental work. Please expand the abstract.
- Introduction is too short and must be expanded, it does not critically discuss past literature and problem in hand.
- Please consider reporting on studies related to your work from mdpi journals.
- Please add a list of nomenclatures and abbreviations at the end of the manuscript for all the Greek symbols and letters used in this study.
- The materials and methods section lacks any graphical images which shows test setup, test equipment and some samples of the analyse wires? This is an experimental study and authors should provide sufficient graphical information for the readers to better understand their work and what was done in it.
- The authors are strongly advised to check the following site: https://www.mdpi.com/journal/metals/instructions for article format according to the journal standards.
- The author should adhere to the manuscript format guidelines of Metals journal. For example, the sub-sectioning using letters is discouraged. Please use 1.1 or 1.1.1 instead of using letters.
- Figure 2 is not clear, please enlarge and improve resolution.
- Table 1 is inserted as an image which is not advised, please use proper table format when add your data in tables.
- Combine figures 3-5 into one larger figure (recommended).
- The authors are strongly encouraged to move all the formulas in the manuscript in an appendix instead of presenting it in the results and discussion section and instead just refer to it from the appendix.
- Table 3 is better presented in a bar chart graph instead of a table, it can quickly give info to the readers about the trends in your results.
- Combine figures 7 and 8 into one figure
- Line 429-430 “are sufficiently small so that each curve could be considered as quasi-stationary state” why? Any explanation for this phenomena? Also what about past studies, did they report similar trends or different from yours, in either way please discuss and support with references.
- Combine figure 9-13 into one larger figure (recommended)
- Line 565-566 please support this claim with a reference(s) if any
- Figure 14 is not clear, please enrlage and enhance resolution.
- Some of the results are merely described and is limited to comparing the experimental observation and describing results. The authors are encouraged to include a more detailed results and discussion section and critically discuss the observations from this investigation with existing literature.
- Conclusion can be expanded or perhaps consider using bullet points (1-2 bullet points) from each of the subsections.
Author Response
We thank you for giving us the opportunity to revise our manuscript “Aqueous dried extract of Skytanthus acutus as corrosion inhibitor of carbon steel in neutral chloride solutions”, for publication in Metals journal. We want to express our appreciation for taking the time and effort necessary to provide an insightful guidance. We hope that these revisions improve the paper such that you now deem it worthy of publication in Metals journal.
Reviewer 2
Thank you for submitting your paper. The work done here draws attention to a significant subject in characterisation of carbon steel. I have found the paper to be interesting. However, several issues need to be addressed properly before the paper is being considered for publication. My comments including major and minor concerns are given below:
- Please consider reviewing the abstract and highlight the novelty, major findings, and conclusions. I suggest reorganizing the abstract, highlighting the novelties introduced. The abstract should contain answers to the following questions:
What problem was studied and why is it important?
What methods were used?
What conclusions can be drawn from the results? (Please provide specific results and not generic ones). Please use numbers or % terms to clearly shows us the results in your experimental work. Please expand the abstract.
A new abstract version was included in the manuscript
- Introduction is too short and must be expanded, it does not critically discuss past literature and problem in hand. Please consider reporting on studies related to your work from mdpi journals.
A more complete introduction was included in the manuscript
3. Please add a list of nomenclatures and abbreviations at the end of the manuscript for all the Greek symbols and letters used in this study.
A list of nomenclatures and abbreviations were included
4. The materials and methods section lacks any graphical images which shows test setup, test equipment and some samples of the analyze wires? This is an experimental study and authors should provide sufficient graphical information for the readers to better understand their work and what was done in it.
All equipment used in this research except for electrodes that are represented in Fig. 1 is standard and normally is implicitly known by readers.
To facilitate the experimental methodology comprehension a graphical representation of the sequence of steps followed in our research was included in the manuscript.
5. The authors are strongly advised to check the following site: https://www.mdpi.com/journal/metals/instructions for article format according to the journal standards.
The manuscript was adjusted accordingly
6. The author should adhere to the manuscript format guidelines of Metals journal. For example, the sub-sectioning using letters is discouraged. Please use 1.1 or 1.1.1 instead of using letters.
Numbers were used for sub sectioning
7. Figure 2 is not clear, please enlarge and improve resolution.
An improved resolution figure was included
8. Table 1 is inserted as an image which is not advised, please use proper table format when add your data in tables.
Table 1 was inserted properly
9. Combine figures 3-5 into one larger figure (recommended).
A combined graph was inserted in the manuscript
10. The authors are strongly encouraged to move all the formulas in the manuscript in an appendix instead of presenting it in the results and discussion section and instead just refer to it from the appendix.
Most formulas were moved to Appendix 1.
11. Table 3 is better presented in a bar chart graph instead of a table, it can quickly give info to the readers about the trends in your results.
icorr and IE values from Tables 2, 5, 6 and 9 were presented in a single bar graph.
12. Combine figures 7 and 8 into one figure
Figs 6 and 7 merged into a one Fig.
13. Line 429-430 “are sufficiently small so that each curve could be considered as quasi-stationary state” why? Any explanation for this phenomena? Also what about past studies, did they report similar trends or different from yours, in either way please discuss and support with references.
Thanks very much for your remark. The sentence is somewhat confusing because do not express the intended idea.
For improved clarity the sentence in the manuscript was changed as” An important requirement for the validity of the kinetic expressions for the partial electrochemical reactions described above in section 2.4 is that each curve could be considered as quasi-stationary state. This assumption applies for the present study and it can be verified as a negligible difference between two consecutive LSV measurements in an electrochemical system”
14. Combine figure 9-13 into one larger figure (recommended).
Several figures pertaining to a similar experimental conditions were combined.
15. Line 565-566 please support this claim with a reference(s) if any.
A reference was added
16. Figure 14 is not clear, please enlarge and enhance resolution.
A better-quality resolution figure was included.
17. Some of the results are merely described and is limited to comparing the experimental observation and describing results. The authors are encouraged to include a more detailed results and discussion section and critically discuss the observations from this investigation with existing literature.
An enhanced discussion is now presented
18. Conclusion can be expanded or perhaps consider using bullet points (1-2 bullet points) from each of the subsections.
Conclusions were improved

Reviewer 3 Report
- For carbon dioxide and hydrogen in Table 1, please note the subscripts.
- Can the quality of the Figs in the paper be further improved? It should be possible to make higher-quality Figs.
- Please also pay attention to the upper and lower indices in Table 2.
- Please provide the corresponding Bode diagram of Nyquist diagram in Figs7,12,13 to judge the degree of fitting of equivalent circuit. In addition, the illustration of equivalent circuit in the Figsis not very clear. In general, the dimensions of horizontal and vertical coordinates of Nyquist graphs are consistent, and the vertical coordinates are negative.
Author Response
We thank you for giving us the opportunity to revise our manuscript “Aqueous dried extract of Skytanthus acutus as corrosion inhibitor of carbon steel in neutral chloride solutions”, for publication in Metals journal. We want to express our appreciation for taking the time and effort necessary to provide an insightful guidance. We hope that these revisions improve the paper such that you now deem it worthy of publication in Metals journal.
Reviewer 3
1. For carbon dioxide and hydrogen in Table 1, please note the subscripts.
done
2. Can the quality of the Figs in the paper be further improved? It should be possible to make higher-quality Figs.
Better quality figures were included in the manuscript
3. Please also pay attention to the upper and lower indices in Table 2.
All indices were upgraded
4. Please provide the corresponding Bode diagram of Nyquist diagram in Figs7,12,13 to judge the degree of fitting of equivalent circuit. In addition, the illustration of equivalent circuit in the Figs is not very clear. In general, the dimensions of horizontal and vertical coordinates of Nyquist graphs are consistent, and the vertical coordinates are negative.
Complete EIS diagrams are now included
Round 2
Reviewer 1 Report
I accept the introduced corrections.
Reviewer 2 Report
All questions answered and paper can be accepted
Reviewer 3 Report
The manuscript can be published in present form.